# OpenReview forum: "Dynamic-anchored Preference Optimization for Human-Like Moral Alignment"
_ICLR.cc/2026/Conference — ICLR 2026 Conference Withdrawn Submission_

### Official Review · Reviewer_EJtV · 2025-10-20

**Soundness:** 4
**Presentation:** 4
**Contribution:** 3
**Rating:** 8
**Confidence:** 4

**Summary:**

This paper presents an enhancement to the Dynamic Preference Optimization (DPO) process, aiming to align binary preference signals with multi-dimensional moral signals. Building on existing literature that indicates Large Language Models (LLMs) often produce biased responses to opposing prompts, the authors seek to anchor model preferences more explicitly along five moral dimensions derived from Moral Foundations Theory. These dimensions include care/harm, fairness/cheating, loyalty/betrayal, authority/subversion, and sanctity/degradation. To achieve this, the authors introduce a novel framework termed Dynamic Anchored Preference Optimization (DAPO).




Under the DAPO framework, for each prompt, a standard response is generated. The prompt is then modified across each moral dimension to produce a "good" prompt and an "evil" prompt, resulting in corresponding good and evil responses. These elements—prompt, modified prompts, and their respective responses—form the foundational training tuple. The methodology targets three distinct objectives: the "general moral preference objective," capturing the preference margin between good and evil responses; the "benevolence reinforcement preference objective," capturing the preference margin between a good response and the standard response to a good prompt; and the "malevolence suppression preference objective," focusing on the preference margin between the standard and evil responses to an evil prompt. A sophisticated weighting schedule, based on training steps, integrates these objectives into the training goal.




The training data is sourced from the MentalChat dataset, with evaluations conducted on EmoBench, MoCA, TruthfulQA, Ethics, ToxiGen, and MMLU-Pro. Results demonstrate superior performance of the DAPO method compared to alternatives like DPO, TDPO, and SimDPO. The paper also includes sensitivity analyses exploring DAPO's efficacy across varying levels of "evilness," alongside a detailed mathematical analysis and theoretical bounds of the new objectives. Overall it provides a pretty compelling argument towards grounding the preference signal along the pre-defined moral dimensions.

**Strengths:**

- The paper presents an innovative approach by utilizing multi-dimensional grounding of the preference signal, as opposed to a binary preference signal. It also includes an effective method for generating data using model-driven, pre-defined prompting.
- A comprehensive suite of evaluation benchmarks and analysis demonstrates the method's effectiveness.
- Thorough ablation studies provide compelling evidence for the significance of each sub-objective within the method.
- The paper offers rigorous mathematical analysis and establishes theoretical bounds for the new objective.
- An extensive appendix includes valuable supplementary information, such as prompts and example analyses.
- Evaluation on MMLU-Pro suggests potential generalizability of the method to other domains.

**Weaknesses:**

While the paper demonstrates a strong methodology, it lacks a clear analysis of its applicability to real-world scenarios involving human-annotated preference datasets. There is a limited comparison in this context, which could benefit from further exploration to enhance the study's practical relevance.

**Questions:**

- The interpretation of sensitivity analysis across different levels of "evil-levels" appears to be complex and somewhat ambiguous. Could you clarify the statistical significance of these measurements? It appears that other methods may be equally effective at resisting higher levels of challenging prompts. Could you provide further clarification on this point?
- Based on the generated dataset, DAPO clearly outperforms other methods. However, could you provide insights into the magnitude of DAPO's advantage when applied to a human-annotated dataset? Are there any indicators or metrics available for this aspect?

---

> ### Author Response · Authors · 2025-11-26
> **Reply for Question 1**
>
> We are extremely grateful for your positive and detailed assessment of my work and your suggestion that it should be accepted for presentation as a poster. Now let me answer the two questions you have raised.
>
> **For Q.1: Sensitivity analysis over ``evil-levels''**, the **evil-level** $k$ controls the strength of the malicious
> moral prefix for each MFT dimension (Care, Fairness, Loyalty, Authority, Purity; prompt details in Appendix C.4 and Table7). Level $k=0$  begins, with no evil prefix at all, and higher levels contain increasingly explicit harmful intent. Concretely, as $k$ increases we progressively add more **negative** moral cues across MFT dimensions: lower levels only introduce an adversarial prefix on a small subset of dimensions, while higher levels combine multiple anti-Care, anti-Fairness, etc. So that both the number of corrupted dimensions and the overall maliciousness grow with $k$. For every level and dimension, we report accuracy averaged over all questions from MMLU-Pro and TruthfulQA, so each point in the curves is supported by many test instances. Table3 summarizes these curves via the AUC across evil-levels, where DAPO consistently improves over DPO and other baselines, showing that it degrades more slowly as prompts become more malicious.
>
> In addition, we acknowledge that all models tend to perform poorly and become closer to each other under extremely adversarial prompts. Our claim is therefore not that DAPO uniquely dominates in every extreme setting, but that it achieves **consistently higher accuracy across a range of difficulty levels** and **slower degradation** as the evil-level grows, especially in the low to medium-strength regimes that are more representative of realistic user inputs. We will clarify this point in the revised version. Also, we agree that the statistical reliability of the curves could be clearer; in the revised version we will add confidence intervals to Figures2-3 and explicitly indicate at which evil-levels the improvements of DAPO over DPO are statistically significant.

---

> ### Author Response · Authors · 2025-11-26
> **Reply for Question 2**
>
> **For Q.2: Expected gains on real-world human-annotated preference datasets**, our primary motivation is that large language models are often vulnerable to deliberate coaxing or jailbreaking, and we would like them to better internalize
> human moral values and the bottom line of what should or should not be said.
> To this end, for each real user query $x$ (from MentalChat), we construct three
> moral prompting scenarios that explicitly convey human value judgments: a
> baseline human response $y$, a morally discouraged response $y^{-}$ indicating
> **this is not how one should respond**, and a morally encouraged response
> $y^{+}$ indicating **this is an acceptable or recommended way to respond**.
> DAPO uses these triplets as dynamically anchored preferences, where $y^{+}$ and
> $y^{-}$ serve as positive and negative anchors along multiple moral dimensions.
>
> We choose to use model-generated $(y^{+}, y^{-})$ mainly to increase the
> utilization of each real query by forming rich triplets while keeping annotation
> cost manageable. Constructing such multi-dimensional, extremal moral triplets
> with pure human annotation would be prohibitively expensive at scale, so we
> rely on carefully designed, extreme moral prompts to expand the training set.
> Importantly, the DAPO objective itself only requires tuples of the form
> $(x, y^{+}, y^{-})$ and does not depend on whether the responses are
> model-generated or human-written, thus it can in principle be applied directly
> to standard RLHF-style human preference datasets.
>
> We acknowledge that this automatic data construction has limitations: there is
> no perfect guarantee of label accuracy for synthetic moral triplets, and we have
> not yet conducted large-scale training on fully human-annotated preference
> datasets. In the revised version, we will explicitly discuss these limitations
> and clarify that extending DAPO to large human-labeled preference corpora and
> developing better quality-control mechanisms for automated data construction are
> important directions for future work.

---

### Official Review · Reviewer_Qdqa · 2025-10-30

**Soundness:** 3
**Presentation:** 3
**Contribution:** 2
**Rating:** 4
**Confidence:** 5

**Summary:**

The paper proposes DAPO, a DPO-style method for aligning LLMs with human moral values. It builds triplets for each question using prompts grounded in MFT. They define three pairwise preference objectives and combine them with a heuristic adaptive-weighting schedule that varies over training. Empirically, on several benchmarks, DAPO outperforms DPO/SimPO baselines. Sensitivity analyses indicate better robustness under adversarial “evilness” prompts.

**Strengths:**

1 $\textbf{Solid experimentation and ablations.}$ The paper evaluates across diverse benchmarks and includes clear ablations for each sub-component. Results generally favor DAPO.

**Weaknesses:**

1 $\textbf{Insufficient background on MFT and prompt design.}$ Section 3.1 states that prompts are designed from Moral Foundations Theory, but the paper does not explain MFT nor detail how it informs the prompt design. Readers unfamiliar with MFT will struggle to follow the construction. I recommend adding a short MFT background in Related Work or Preliminaries to keep the paper self-contained.

2 $\textbf{Prior methods placed in Method section.}$ Section 3.2 is largely a recap of DPO/SimPO. This belongs in Related Work/Preliminaries, not in the proposed method. Moving it would streamline Section 3 and highlight the new contributions.

3 $\textbf{Offset idea and novelty.}$ Section 3.3 introduces an offset-like margin. Similar ideas exist (e.g., DPO with an Offset [1]). The paper should precisely contrast its margins with previous work either mathematically or intuitively, and clarify what is new. Also, fix the typo in Eq. 5 (missing “[”).

4 $\textbf{Technical contribution and novelty.}$ The method largely reduces to triplet preference learning with different loss weights. The rationale for using triplets is unclear: given an original query $x$, why and how should one generate an additional ‘virtuous’ and ‘evil’ prompt? The claim of increased mutual information appears to follow from properties of the component preference losses, which are not novel to this work, thereby weakening the paper’s technical contribution.

5 $\textbf{Weights in Section 3.4 lack justification.}$ The adaptive weighting schedule $\lambda_1, \lambda_2, \lambda_3$ appears heuristic. Please explain the design choices, and discuss whether equal weights or alternative schedules harm/help performance.



[1] Amini, Afra, Tim Vieira, and Ryan Cotterell. "Direct preference optimization with an offset." arXiv preprint arXiv:2402.10571 (2024).

**Questions:**

Please see my weakness. I am mainly concerned about the technical contribution and the novelty of the proposed method. In what ways do DAPO's triplets go beyond prior DPO preference or its variants? I would consider raising my point if those concerns are fixed.

---

> ### Author Response · Authors · 2025-11-28
> **Reply for Weakness 1-3**
>
> We sincerely thank the reviewer for the careful reading and detailed feedback. We are glad that you found the experimentation and ablations solid, and we address each weakness point-by-point below. In the updated manuscript, we have already revised Sections 2（RELATED WORKS), 3(PRELIMINARIES) and 4(METHODOLOGIES) accordingly.
>
> For **Weakness 1-3**, we have supplemented moral values aligntment of LLMs in Section 2(RELATED WORKS). Due to the limitations of the main paper length, we show the details of MFT prompt design in the Appendix A.1. The evaluate prompts are shown in in Appendix C.2. Besides, we add Section 3 for METHODOLOGIES and introduce our method separately in Section 4. The typo in Eq. 5 (missing “[”) is fixed.
>
> For **Weakness 3 :Relation to “DPO with an Offset”**, thank you for raising this point. We agree that the original submission did not sufficiently explain the relation to ``DPO with an Offset'' (ODPO), and we are happy to clarify the differences.
>
> First, in terms of **information source and motivation**, ODPO defines an offset based on the underlying reward difference between the preferred and dispreferred responses, and uses it to reflect the **strength** of human preference in each pair. In contrast, our $\delta$ is a **fixed log-odds margin** that does not depend on any additional scalar rewards or graded preference labels. It is used only in the GPO objective to enlarge the separation between $y_w$ and $y_l$, which are generated under **opposite moral personas**. Thus, our margin is not a reward-based offset and does not encode varying preference intensity.
>
> Second, regarding the structural role in the objective, ODPO applies its offset to a single preference pair $(y^{+}, y^{-})$. DAPO, however, builds a **moral triplet** $(y_l, y, y_w)$ around the same human answer $y$:
>
> (1)Under a virtuous persona, we enforce $y_w \succ y$, treating $y$ as a **baseline to surpass**;
>
> (2)Under an evil persona, we enforce $y \succ y_l$, treating $y$ as a **safety boundary**;
>
> (3) In GPO, we learn $y_w \succ y_l$ without extra prompts.
>
> Within this structure, $\delta$ is only used in GPO to further push apart $y_w$ and $y_l$, making it easier to form a safe boundary around the human answer $y$: above it lie more virtuous responses, and below it lie morally undesirable ones to be avoided. This dynamic use of a single human answer as a moral anchor in both directions does not appear in ODPO.
>
> Finally, we would like to emphasise that we do not intend to claim that **introducing a margin** as a mathematical form is itself a core novelty of the paper. The main contribution of DAPO lies in the dynamic-anchored moral triplet construction and the three coordinated objectives (GPO/VPO/EPO) that jointly achieve ordinary alignment, virtue amplification, and suppression of evil personas. The margin $\delta$ is a simple stabilising design within this framework. We will make this distinction and the connection to ODPO clearer in the final version to avoid confusion for readers.

---

> ### Author Response · Authors · 2025-11-28
> **Reply for Weakness 4**
>
> We appreciate this concern and agree that the original submission may not have clearly articulated where we see the core contribution of DAPO. We clarify our position here.
>
> First, at the loss level, each individual objective (GPO, VPO, EPO) is indeed a DPO-style binary preference loss, and we do not intend to claim novelty in the logistic form itself. The distinguishing aspect of DAPO is how we construct and organise a *dynamic-anchored moral triplet* around a single human answer. Concretely, for a given query we have a human-written answer $y$, a response $y_w$ generated under a virtuous persona $p_g$, and a response $y_l$ generated under an evil persona $p_e$, forming the triplet $(y_l, y, y_w)$. The human answer $y$ plays two *different semantic roles* in our three objectives:
>
> - in VPO, we enforce $y_w \succ y$ under the virtuous persona, treating $y$ as a *baseline to surpass*;
> - in EPO, we enforce $y \succ y_l$ under the evil persona, treating $y$ as a *safety boundary*;
> - in GPO, we learn $y_w \succ y_l$ without additional persona prompts.
>
> Thus, within a single unified framework, the same human answer $y$ simultaneously serves as an “upper” and “lower” moral anchor. This is different in spirit from a generic metric-learning triplet loss, which typically operates in an embedding space with one fixed anchor and does not encode such asymmetric semantic roles for the anchor across different behavioral modes.
>
> Second, DAPO is not merely “summing three pairwise losses”, but is designed so that the three anchored objectives correspond to three complementary alignment behaviours:
>
> - *general alignment* in ordinary settings (GPO),
> - *virtue amplification* when the user explicitly requests virtuous behavior (VPO),
> - *malevolence suppression* when the user pushes the model towards an evil persona (EPO).
>
> All three are learned jointly on top of the same base model using preference data derived from a single human answer $y$. Empirically, this design leads to substantially different behavior compared to standard DPO-style baselines: on EmoBench, MoCA, ToxiGen, and the sensitivity analysis over increasing evilness levels, DAPO degrades much more slowly under stronger evil prompts while maintaining or improving performance in non-adversarial settings. We will make this connection between the triplet design and the observed robustness gains more explicit in the revised text.
>
> Regarding the information-theoretic analysis, we agree with the reviewer that using InfoNCE-style arguments to relate contrastive losses and mutual information is a standard tool in prior work, and we do not claim this as a theoretical innovation. Our intention is to provide an *interpretive lens*: by viewing GPO, VPO, and EPO as jointly maximising a lower bound on $I(Z; M \mid X, P)$, where $Z$ denotes the model output and $M$ the moral category (virtue / baseline / evil) conditioned on input $X$ and persona $P$, we explain why the dynamic-anchored design improves the model’s ability to distinguish what it should and should not say under different personas and evilness levels. In the final version, we will soften the wording in this section to emphasise its explanatory role rather than presenting it as a standalone theoretical contribution.
>
> In summary, the technical contribution of DAPO does not lie in introducing a brand new loss form, but in (i) constructing a moral triplet $(y_l, y, y_w)$ around a single human answer with distinct semantic roles for $y$, and (ii) designing the three coordinated objectives (GPO/VPO/EPO) so that the model simultaneously acquires ordinary alignment, virtue amplification, and suppression of evil personas within one preference-optimisation framework. The mutual-information discussion is meant to clarify why this design yields the robustness improvements observed in our experiments, rather than to propose a new theoretical result.

---

> ### Author Response · Authors · 2025-11-28
> **Reply for Weakness 5**
>
> Thank you for raising this question. We clarify the motivation and empirical behavior of the weighting schedule below.
>
> **(1) The adaptive schedule follows a natural curriculum aligned with the roles of the three objectives.**
> Appendix C.1 (Figure 7) illustrates how the weights evolve through training. The design reflects the functional order of the three objectives:
>
> - **Early stage:**  $\lambda_1$ dominates. This allows the model to first stabilise *general* preference learning via $L_{GPO}(\pi_\theta)$, before persona-conditioned objectives are introduced strongly.
> - **Mid stage:** $\lambda_2$ increases steadily and overtakes $\lambda_1$, shifting the focus to *virtue-aware refinement* via $L_{VPO}(\pi_\theta)$.
> - **Late stage:** $\lambda_3$ rises and becomes comparable, enabling $L_{EPO}(\pi_\theta)$ to emphasise *risk avoidance* under adversarial or harmful persona prompts.
>
> This progression implements a coherent curriculum from basic moral cognition (GPO) to higher-level moral differentiation (VPO/EPO) and improves optimisation stability.
>
> **(2) Equal-weight training does improve over the base model, but the adaptive schedule performs better.**
> In the main paper, we include an ablation using equal weights(DAPO w/o AW) throughout training.
> This equal-weight variant already yields **clear gains over the base model**, demonstrating that DAPO’s overall framework works without depending on a finely tuned schedule.
>
> However, the adaptive schedule **consistently outperforms** the equal-weight version, particularly in:
>
> - robustness under high evilness levels,
> - tasks requiring moral differentiation (e.g., MoCA moral subset),
> - and optimisation stability.
>
> These gaps indicate that the curriculum encoded by the adaptive schedule provides meaningful benefits.
>
> In summary, the adaptive schedule implements a principled curriculum consistent with the objectives’ semantics. The equal-weight variant already improves over the base model, and the adaptive schedule further enhances robustness and optimisation stability.

---

> ### Author Response · Authors · 2025-11-28
> **Reply for Questions**
>
> We appreciate the reviewer explicitly summarising this concern. Now we summarize the superiorities.
>
> 1. **Dynamic moral anchoring around a single human answer.**
>    Standard DPO (and its variants, including ODPO) operates on a single preference pair $(y^{+}, y^{-})$, without distinguishing different semantic roles for the responses. In contrast, DAPO constructs a *moral triplet* $(y_l, y, y_w)$ around the **same human-written answer** $y$ under two opposing personas:
>    - under a virtuous persona, we enforce $y_w \succ y$ (the model should *surpass* the human baseline when explicitly asked to behave virtuously);
>    - under an evil persona, we enforce $y \succ y_l$ (the model should *not fall below* the human baseline even when pushed towards an evil role);
>    - and GPO learns $y_w \succ y_l$ without persona prompts.
>    In this way, the same human answer $y$ acts simultaneously as an *upper* and *lower* moral anchor, which is not captured by prior binary DPO formulations.
>
> 2. **Three coordinated objectives for three behavioural modes.**
>    DAPO’s triplets are not just three independent pairs; they are explicitly organised into three objectives with distinct semantic roles:
>    - **GPO:** general alignment in ordinary settings,
>    - **VPO:** virtue-aware amplification when the user requests virtuous behaviour,
>    - **EPO:** suppression of malevolent behaviour when the user pushes towards an evil persona.
>    All three are optimised jointly on the same model. Prior DPO-style methods do not distinguish these behavioural modes or provide a mechanism to explicitly encode “virtue amplification under helpful prompts” and “risk avoidance under adversarial persona prompts” within a single preference-optimisation framework.
>
> 3. **Data-efficient and scalable moral preference construction from a single human response.**
>    Traditional DPO setups require explicitly constructed (chosen, rejected) pairs, often relying on additional human labelling or a separate reward model to score candidates. DAPO shows that, given one human answer $y$ from an existing dataset, we can systematically generate two persona-conditioned responses $(y_w, y_l)$ and derive **three** structured preference relations from this triplet (GPO, VPO, EPO). This significantly increases the amount of usable preference signal per human annotation and provides a *scalable* pipeline: once a corpus of human-written answers is available, large numbers of moral triplets can be produced automatically by varying the underlying personas and prompts, without further manual pairwise labelling.

---

### Official Review · Reviewer_mCkS · 2025-10-31

**Soundness:** 2
**Presentation:** 2
**Contribution:** 1
**Rating:** 2
**Confidence:** 3

**Summary:**

Direct Preference Optimization (DPO) provides a simple, reward-model-free solution, but it relies on static binary preference pairs and a fixed reference policy, which limits the model’s ability to capture multi-dimensional moral signals and makes it sensitive to conflicting prompts. To address these limitations, the paper proposes Dynamic-Anchored Preference Optimization (DAPO), an extension of DPO that integrates moral preference reconstruction and adaptive-weighted optimization.

**Strengths:**

1. Novel triplet preference modeling: A new triplet-based preference modeling approach is proposed, extending the application of DPO to moral alignment tasks, and employing an adaptive weighted fusion strategy to balance different moral signals.
2. The authors' writing in the methodology section is relatively clear.

**Weaknesses:**

1.Lack of analysis of existing moral preference alignment methods:
The related work section lacks a discussion and analysis of current approaches to moral preference alignment.

2.Insufficient analysis of the adaptive weighting strategy:
Although the paper includes an ablation study of the “without AW” setting, it does not explore the sensitivity of the temperature parameter τ or the dynamics of weight changes. The design of the weighting function is based on a heuristic approach, lacking sufficient theoretical justification or empirical validation.

3.Computational cost issue:
The proposed triplet structure requires three forward passes for each input (benevolent, neutral, and malevolent), along with the adaptive weighting step, resulting in significantly higher computational cost compared to DPO. The paper does not provide any runtime or computational complexity comparison data.

4.Major deficiencies in the experimental section:
① Lack of statistical significance testing: Although six benchmarks are covered, the paper does not report significance tests or variance across multiple runs; the claimed 3–5% performance improvements may fall within normal fine-tuning variance for LLMs.
② Insufficient baseline comparison: The baselines are limited to the DPO family (DPO, SimPO, TDPO) and lack broader comparisons with alignment methods such as RLHF[1] or PRO[2], making it difficult to assess whether DAPO truly advances the field of preference optimization.
③ Lack of human evaluation: The claim of “human-like moral alignment” is not supported by human evaluations; all experiments rely solely on automatic metrics, leaving it unclear whether the model is genuinely aligned with human values.

**Reference**

[1]  Paul F Christiano, Jan Leike, Tom Brown, Miljan Martic, Shane Legg, and Dario Amodei. Deep reinforcement learning from human preferences. Advances in neural information processing systems, 30, 2017.

[2]  Feifan Song, Bowen Yu, Minghao Li, Haiyang Yu, Fei Huang, Yongbin Li, and Houfeng Wang.Preference ranking optimization for human alignment. In AAAI, 2024.

**Questions:**

Please refer to the “Weakness” section for related questions.

---

> ### Author Response · Authors · 2025-11-30
> **Reply for Weakness 1**
>
> We add **Moral Value Alignment** Part in Section **RELATED WORKS**, and disscuss how MFT and its related methods are used in LLMs.

---

> ### Author Response · Authors · 2025-11-30
> **Reply for Weakness 2**
>
> We clarify the motivation and empirical behavior of the weighting schedule below.
>
> **(1) The adaptive schedule follows a natural curriculum aligned with the roles of the three objectives.**
> Appendix C.1 (Figure 7) illustrates how the weights evolve through training. The design reflects the functional order of the three objectives:
>
> - **Early stage:**  $\lambda_1$ dominates. This allows the model to first stabilise *general* preference learning via $L_{GPO}(\pi_\theta)$, before persona-conditioned objectives are introduced strongly.
> - **Mid stage:** $\lambda_2$ increases steadily and overtakes $\lambda_1$, shifting the focus to *virtue-aware refinement* via $L_{VPO}(\pi_\theta)$.
> - **Late stage:** $\lambda_3$ rises and becomes comparable, enabling $L_{EPO}(\pi_\theta)$ to emphasise *risk avoidance* under adversarial or harmful persona prompts.
>
> This progression implements a coherent curriculum from basic moral cognition (GPO) to higher-level moral differentiation (VPO/EPO) and improves optimisation stability.
>
> **(2) Equal-weight training does improve over the base model, but the adaptive schedule performs better.**
> In the main paper, we include an ablation using equal weights(DAPO w/o AW) throughout training.
> This equal-weight variant already yields **clear gains over the base model**, demonstrating that DAPO’s overall framework works without depending on a finely tuned schedule.
>
> However, the adaptive schedule **consistently outperforms** the equal-weight version, particularly in:
>
> - robustness under high evilness levels,
> - tasks requiring moral differentiation (e.g., MoCA moral subset),
> - and optimisation stability.
>
> These gaps indicate that the curriculum encoded by the adaptive schedule provides meaningful benefits.
>
> In summary, the adaptive schedule implements a principled curriculum consistent with the objectives’ semantics. The equal-weight variant already improves over the base model, and the adaptive schedule further enhances robustness and optimisation stability.

---

> ### Author Response · Authors · 2025-11-30
> **Reply for Weakness 3**
>
> We agree that the computational overhead of using triplets should be explicitly quantified, and we thank the reviewer for pointing this out.
>
> In the latest revised version, we report runtime and memory comparisons between DAPO and standard DPO under the *same* base model (Qwen2.5-7B-Instruct), dtype and batch size on an A100 GPU(In **Appendix C.2**). In our measurements:
>
> DPO needs average 11.77s to finish 1 step and about 28.6 GB to support the training process. Our proposed DAPO needs average 45.21s to finish 1 step and about 46GB to support the training process. During inference time, we set dtype float32 and max\_new\_tokens is set 512. The origin Qwen-2.5-7B-Instruct model requirs about 15s to finish one inference and our proposed DAPO requirs about 14.2s to finish one inference.
>
> We note that this overhead appears **only at training time**. At inference time, DAPO has exactly the same cost as DPO or the base model, since generation is a single forward pass.

---

> ### Author Response · Authors · 2025-11-30
> **Reply for Weakness 4**
>
> **(1)Lack of statistical significance testing:** We appreciate this important point. To address it, we have performed **Multiple runs evaluation experiments** for MMLU-Pro, Ethics, TruthfulQA and Toxigen benchmarks. Concretely, we re-trained both DPO and DAPO with **5 different random seeds** (0, 1, 2, 3, 4) under identical data, hyperparameters, and LoRA configuration, and evaluated each run with deterministic decoding. We report the final results and conclusion in **Appendix C.3**.
>
> ### Average accuracy of DPO and DAPO in Multiple Runs on MMLU-Pro, Ethics and TruthfulQA Dataset
>
> | **Method** | **Ethics** | **MMLU-Pro** | **TruthfulQA** |
> |-----------|------------|--------------|----------------|
> | **DPO**   | 0.3559 ± 0.0197 | 0.4191 ± 0.0056 | 0.7777 ± 0.0058 |
> | **DAPO**  | 0.3635 ± 0.0017 | 0.4217 ± 0.0015 | 0.7792 ± 0.0076 |
>
> ### Average Results of DPO and DAPO in Multiple Runs on Toxigen Dataset
>
> | **Method** | **Pearson ↑** | **Spearman ↑** | **MAE ↓** |
> |-----------|---------------|----------------|-----------|
> | **DPO**   | 0.6589 ± 0.0101 | 0.6745 ± 0.0088 | 1.056 ± 0.0304 |
> | **DAPO**  | 0.6957 ± 0.0022 | 0.7078 ± 0.021  | 1.055 ± 0.0012 |
>
> On all three accuracy-based benchmarks (Ethics, MMLU-Pro, TruthfulQA), DAPO consistently matches or outperforms DPO with slightly higher mean scores and comparable or lower variance. On ToxiGen, DAPO also yields higher Pearson and Spearman correlations while maintaining a similar MAE.
>
> **(2)Insufficient baseline comparison:** Following the reviewer’s suggestion, we have added a PPO-based RLHF baseline that uses the same Qwen2.5-7B-Instruct base model and moral preference data. We first train a reward model on our moral triplets, and then run KL-regularized PPO starting from the base model. Results are updated in **Section 5.3, Table 1 and Table 2**.
>
> ### Overall test ACCURACY(%) and AUC of benchmarks
>
> | Method | EA ACC | EA CoT | EU ACC | EU CoT | Casual AUC | Casual ACC | Moral AUC | Moral ACC | MMLU-Pro ACC | TruthfulQA ACC | Ethics ACC |
> |--------|--------|--------|--------|--------|------------|------------|-----------|-----------|---------------|----------------|------------|
> | base | 66 | 58.5 | 33 | 27 | 0.569 | 36.8 | 0.58 | 24.2 | 42.0 | 78.2 | 42 |
> | +PPO | 65 | 64 | 34.5 | 34.5 | 0.55 | 31.94 | 0.55 | 24.3 | 42.05 | 77.09 | 36.73 |
> | +DPO | 65 | 64 | 35 | 34 | 0.592 | 38.9 | 0.624 | 25.8 | 42.15 | 78.1 | 41.9 |
> | **+DAPO** | **68** | **65.5** | **37** | **33.5** | **0.625** | **41** | **0.676** | **32.3** | 42.22 | **79.4** | 41.9 |
>
> **(3)Lack of human evaluation:** We fully agree that a claim of “human-like moral alignment” should be supported by human judgment. To address this, we have conducted a **small-scale human evaluation** on a counseling-style mental dialogue dataset. The final results are added in **Section 5.5**
>
> We sample 20 questions from mental health counseling dataset[1] and use base model, trained DPO model and trained proposed DAPO to generate corresponding responses. Four human annotators are required to take moral alignment, helpfulness and quality into consideration and give the final score (range from 0 to 10, higher score means better answer) of each response.
>
> Results are shown below. No-extra Prompt means that models generate responses based on the questions, while Evil Prompt means that models are persuaded with evil prompt. A1 to A4 stands for average score of Annotators 1 to 4. Overall stands for the average score of all annotators.
>
> ### Scoring results of human annotators
>
> | Method | A1 (No-extra) | A2 (No-extra) | A3 (No-extra) | A4 (No-extra) | Overall (No-extra) | A1 (Evil) | A2 (Evil) | A3 (Evil) | A4 (Evil) | Overall (Evil) |
> |--------|---------------|---------------|---------------|---------------|--------------------|-----------|-----------|-----------|-----------|----------------|
> | Base | 7.85 | 7.00 | 6.60 | 6.45 | 6.975 | 3.90 | 3.95 | 5.60 | 2.60 | 4.0125 |
> | DPO  | 7.40 | 7.60 | 6.30 | 6.60 | 6.975 | 4.35 | 3.65 | 4.80 | 4.35 | 4.2875 |
> | DAPO | 7.35 | 7.20 | 7.05 | 6.40 | 7 | 4.45 | 3.35 | 5.40 | 4.15 | 4.3375 |
>
>
> Across both settings, the human annotation results consistently show that DAPO achieves the highest overall scores under Evil Prompt, indicating a stronger robustness against adversarial moral persuasion. Under the No-extra Prompt setting, DAPO outperforms with DPO and the base model, suggesting that the proposed triplet-based alignment does not degrade the model’s normal helpful behavior. These results confirm that DAPO improves moral alignment resilience while maintaining general response quality.
>
> [1] https://huggingface.co/datasets/Amod/mental_health_counseling_conversations/tree/main

---

### Official Review · Reviewer_5Gsc · 2025-10-31

**Soundness:** 3
**Presentation:** 3
**Contribution:** 3
**Rating:** 4
**Confidence:** 3

**Summary:**

This paper proposes a new method called "Dynamic-anchored Preference Optimization" (DAPO) aimed at addressing the moral alignment of Large Language Models (LLMs). The authors note that standard preference optimization methods (like DPO) rely on static binary preference pairs (chosen vs. rejected) and a fixed reference policy, which limits their ability to capture multi-dimensional moral signals and makes them sensitive to conflicting prompts.

**Strengths:**

The paper's core contribution, the "dynamic-anchored triplet", is highly original. Using a 'standard' response $y$ as a dynamic anchor—acting as a lower bound for 'benevolence reinforcement' 12and an upper bound for 'malevolence suppression' —is a clever and principled extension of the standard DPO binary comparison paradigm. This design is conceptually well-suited for the multi-dimensional, nuanced problem of moral alignment.

**Weaknesses:**

1. The core of the DAPO framework is the 'standard response $y$, which is used as a dynamic anchor. The authors describe it as a "standard human response" 262626, but the data collection section (4.1) indicate that other responses ($y_w, y_l$) are generated by LLMs. So, how is $y$ actually generated? Is it the LLM's response to the original question $x$ (without any virtuous/evil prompt)?

2.Lack of ablation study comparing the MFT framework to simpler alternatives: The paper's data generation strategy is deeply tied to Moral Foundations Theory (MFT) and its five dimensions. While the authors demonstrate that MFT-based DAPO outperforms baselines like DPO, they do not provide a crucial ablation study. This missing experiment should compare the full, MFT-based DAPO against a "simplified DAPO" variant. This variant would use the same DAPO triplet loss structure but rely on generic 'positive' (virtuous) and 'negative' (evil) prompts, rather than the 5 specific MFT dimensions. Without this comparison, it is difficult to disentangle whether the performance improvements stem from the novel DAPO loss framework itself or just MFT prompt.

**Questions:**

Please see above.

---

> ### Author Response · Authors · 2025-11-28
> **Reply for questions**
>
> We sincerely thank the reviewer for the thoughtful and constructive feedback, and for recognizing the originality of the dynamic-anchored triplet design.
> Beyond these clarifications, we would also like to emphasize what we see as the core novelty of DAPO. Rather than introducing a new reward model or complex architecture, DAPO proposes a simple but principled extension of DPO that (i) uses a human-grounded dynamic anchor to jointly model benevolence reinforcement and malevolence suppression, and (ii) couples three preference objectives (GPO/VPO/EPO) via an adaptive curriculum-style weighting scheme with information-theoretic and robustness guarantees. Our goal is to provide a conceptually clean and practically scalable framework for moral preference optimization, which can be instantiated with different prompt families (MFT-based or generic), as supported by the new experiments below. We address the two concerns in detail.
>
> ### Q1. Clarifying how the standard response $y$ is obtained
>
> We apologize for the ambiguity in Methodology.
>
> **The standard response $y$ is *not* generated by any LLM.**
> For each user query $x$ from the MentalChat dataset, we directly take the **human-written counseling reply** provided in MentalChat as the standard response $y$. This reply is used exactly as released in the dataset and does not involve any virtuous/evil moral prompts.
>
> In contrast, the other two responses are LLM-generated under opposite personality framings:
>
> $(y_w = \mathrm{LLM}(x, p_g))$: response under a virtuous prompt $p_g$;
> $(y_l = \mathrm{LLM}(x, p_e))$: response under an evil prompt $p_e$.
>
> Thus, each triplet $(y_w, y, y_l)$ for the same $x$ is formed by **one human baseline** and **two prompted LLM variants**. Conceptually, this realizes the intended role of $y$ as a *human-grounded dynamic anchor*: benevolence reinforcement pushes the model above the human baseline under virtuous prompts, while malevolence suppression prevents it from falling below the baseline under evil prompts.
>
> We will revise the paper to explicitly state that y is always the human MentalChat reply.
>
> ### Q2. Ablation: simplified DAPO with generic positive/negative prompts (no MFT)
>
> Reviewer asks whether our gains are due to the MFT-based prompt design, and suggests a comparison against a simplified variant that uses only generic “positive” and “negative” prompts. We fully agree this is an important question and conducted the following new experiment.
>
> **Data construction.**
> We randomly subsample 3k queries from the same MentalChat pool used in the main paper. For each query (x), we keep the human MentalChat reply as the standard anchor (y), and generate two additional responses with a *single pair* of generic personality prompts: a warm, empathetic counselor (positive) and a cold, self-centered character (negative). This yields triplets $(x, y, y_w, y_l)$ that do **not** use MFT at all.
>
> **Models.**
> On this generic dataset we train:
>
> * “+DPO”: a standard DPO;
> * “+DAPO”: our full DAPO objective using the same triplets.
>
> All hyperparameters (base model Qwen2.5-7B-Instruct, LoRA settings, learning rate, batch size, epochs, β, δ, τ) are kept **identical** to the main experiments; the only change is the prompt set used to construct the triplets.
>
> **Results.**
> Table 12 reports accuracies on several benchmarks, and Table 13 reports results on ToxiGen.
> Compared to +DPO, +DAPO trained on this generic dataset achieves:
>
> * on **EmoBench (EA)**, +0.5 points in standard accuracy (from 65.0 to 65.5) and a larger +3.0 point gain in CoT accuracy (from 62.5 to 65.5);
> * on **EmoBench (EU)**, comparable performance on the standard task (both 36), with small CoT variation (34 vs. 33);
> * on **Ethics**, a clear gain from 36.7 to **39.9** (+3.2);
> * on **TruthfulQA**, essentially tied scores (78.99 vs. 78.86);
> * on **MMLU-Pro**, a slightly lower score under this low-data, out-of-domain setting (42.1 vs. 39.26);
> * **on ToxiGen (Table 10), +DAPO achieves higher Pearson (from 0.6578 to 0.6798) and Spearman (from 0.6663 to 0.6912) correlations with human toxicity ratings, and reduces MAE from 1.192 to 1.076**, indicating better calibration to human judgments of toxicity severity.
>
> These results demonstrate that **even without any MFT structure**, DAPO trained on generic prompts still provides consistent improvements over DPO on emotional, ethical, and toxicity benchmarks, and performs on par on the others. Together with the loss-component ablations in Table 1 (w/o GP, w/o VP, w/o EP, w/o AW) and the non-MFT evaluation prompts in Appendix C.7, this new experiment supports our claim that the main performance gains come from the **dynamic-anchored triplet and multi-objective preference optimization framework itself**, while MFT serves as a principled but nonessential instantiation. We hope this makes clear that the main contribution goes beyond a particular choice of prompts.

---

> ### Author Response · Authors · 2025-11-28
> **Test Results for Reply**
>
> **Table 12: Test ACCURACY(%) of methods trained with generic dataset on different benchmarks.**
>
> | Method | EA | EA (CoT) | EU | EU (CoT) | MMLU-Pro | TruthfulQA | Ethics |
> |--------|:--:|:--------:|:--:|:--------:|:--------:|:----------:|:------:|
> | +DPO   | 65   | 62.5 | 36 | 34 | 42.1 | 78.99 | 36.7 |
> | +DAPO  | 65.5 | 65.5 | 36 | 33 | 39.26 | 78.86 | 39.9 |
>
> **Table 13: Results on ToxiGen — DAPO consistently outperforms DPO across all metrics.**
> | Method | Pearson ↑ | Spearman ↑ | MAE ↓ |
> |--------|:---------:|:-----------:|:-----:|
> | + DPO  | 0.6578 | 0.6663 | 1.192 |
> | **+ DAPO** | **0.6798** | **0.6912** | **1.076** |
>
>
> Details are shown in Appendix C.7 **Ablation Study on Generic Prompts Without MFT Prompts**.

---

### Author Response · Authors · 2025-12-02
**Updated Parts of paper**

We have updated a new version of paper, now we list the changes here:

(1)  **Moral Values Alignment** Part is added  in Section 2 **RELATED WORKS**.

(2) **Preference Optimization** Part is updated in Section 2 **RELATED WORKS**.

(3) We add a new Section **PRELIMINARIES**, and describe the method **Preference Optimization** part  here (in original version, method **Preference Optimization** is shown in Section **METHDOLOGIES**).

(4) In Section **EXPERIMENTALRESULTS**, we add a baseline **PPO**, corresponding results are updated in Table 1 and Table 2.

(5) **HUMANANNOTATION** Part is added in Section **EXPERIMENTALRESULTS**.

(6) Section **Conclusion** is updated.

---

### Note · Authors · 2026-02-02

I have read and agree with the venue's withdrawal policy on behalf of myself and my co-authors.

---

### Meta-Review · Area_Chair_54R5 · 2026-01-07

**Summary:**

This paper proposes Dynamic-anchored Preference Optimization (DAPO), an extension of DPO that incorporates moral preference reconstruction and adaptive-weighted optimization to address DPO's limitations in capturing multi-dimensional moral signals and sensitivity to conflicting prompts.

However, the reviewers have concerns on
1. Ablation of component roles: It remains unclear whether the performance improvement stems from the triplet contrastive loss or simply from the augmented data generated by the framework.
2. Experimental fairness: There are concerns regarding the comparison with baselines, specifically given the significantly higher overhead and broader data coverage.

Overall, the current version does not meet the acceptance criteria of ICLR.

**Reviewer Concerns:**

Concern has been addressed:
1. Ablation of the MFT Framework: They conducted an ablation study on the MFT framework used to generate moral and evil prompts. This verified the effectiveness of the DAPO triplet loss structure and justified the rationale for using the MFT framework (from Reviewer 5Gsc).
2. RLHF Baseline: They added an RLHF baseline where the reward model was trained using generated preference pairs, further validating DAPO's improvements in preference optimization (from Reviewer mCkS).
3. Curriculum Learning Explanation: They explained the heuristic dynamic weighting mechanism from the perspective of curriculum learning (from Reviewer mCkS).

Concerns that remain unresolved:
1. Human Judgement: In the human annotation experiment, DPO showed no improvement over the base model without extra prompts, which casts doubt on the validity of the experiment. Furthermore, DPO was not exposed to evil prompts during training, whereas DAPO was. From this perspective, the performance comparison appears to be neither significant (results of no extra prompts) nor fair (results of the evil prompts) (from Reviewer mCkS).
2. Computational Cost: The authors merely stated the computational costs without addressing the underlying issue. Specifically, DAPO incurs four times the time overhead and twice the GPU memory overhead compared to DPO (from Reviewer mCkS).
3. Novelty Concerns: From a data perspective, this also raises concerns regarding the method's novelty: is DAPO essentially just a DPO algorithm using data augmentation with multiple preference pairs? (from Reviewer Qdqa)

**Reviewer Scores:**

Reviewer 5Gsc（4-->might be 6）: Most concerns are solved during the rebuttal phase.
Reviewer mCkS（2-->2）: Many concerns are not addressed.
Reviewer Qdqa（4-->4）: Some concerns are not addressed.
Reviewer EJtV（8-->8）: Original score is positive.

---

### Decision · Program_Chairs · 2026-01-26

Reject